# Remotely Powered Two-Wire Cooperative Sensors for Bioimpedance Imaging Wearables

**DOI:** 10.3390/s24185896

**Published:** 2024-09-11

**Authors:** Olivier Chételat, Michaël Rapin, Benjamin Bonnal, André Fivaz, Benjamin Sporrer, James Rosenthal, Josias Wacker

**Affiliations:** 1Medtech Business Unit, Swiss Center for Electronics and Microtechnology (CSEM), Jaquet-Droz 1, 2002 Neuchâtel, Switzerlandandre.fivaz@csem.ch (A.F.); james.rosenthal@csem.ch (J.R.); josias.wacker@csem.ch (J.W.); 2Integrated & Wireless Business Unit, Swiss Center for Electronics and Microtechnology (CSEM), Technopark, Technoparkstrasse 1, 8005 Zürich, Switzerland; benjamin.sporrer@csem.ch

**Keywords:** bioimpedance imaging, electrical impedance tomography (EIT), active electrode, dry electrode, cooperative sensor, wearables, medical device, cardiopulmonary diseases, respiratory diseases, COPD

## Abstract

Bioimpedance imaging aims to generate a 3D map of the resistivity and permittivity of biological tissue from multiple impedance channels measured with electrodes applied to the skin. When the electrodes are distributed around the body (for example, by delineating a cross section of the chest or a limb), bioimpedance imaging is called electrical impedance tomography (EIT) and results in functional 2D images. Conventional EIT systems rely on individually cabling each electrode to master electronics in a star configuration. This approach works well for rack-mounted equipment; however, the bulkiness of the cabling is unsuitable for a wearable system. Previously presented cooperative sensors solve this cabling problem using active (dry) electrodes connected via a two-wire parallel bus. The bus can be implemented with two unshielded wires or even two conductive textile layers, thus replacing the cumbersome wiring of the conventional star arrangement. Prior research demonstrated cooperative sensors for measuring bioimpedances, successfully realizing a measurement reference signal, sensor synchronization, and data transfer though still relying on individual batteries to power the sensors. Subsequent research using cooperative sensors for biopotential measurements proposed a method to remove batteries from the sensors and have the central unit supply power over the two-wire bus. Building from our previous research, this paper presents the application of this method to the measurement of bioimpedances. Two different approaches are discussed, one using discrete, commercially available components, and the other with an application-specific integrated circuit (ASIC). The initial experimental results reveal that both approaches are feasible, but the ASIC approach offers advantages for medical safety, as well as lower power consumption and a smaller size.

## 1. Introduction

Unlike other imaging modalities (e.g., MRI), bioimpedance imaging and electrical impedance tomography (EIT) [1,2,3,4,5,6,7] allow for time-parameterized maps of tissue resistivity and permittivity at a high frame rate. Bioimpedance imaging and EIT are safe and potentially inexpensive. Moreover, they can be made wearable. However, the spatial resolution is quite low (as compared to, e.g., MRI). The resistivity/permittivity maps can be processed further to provide non-invasive and wearable modalities, e.g., for pulmonary artery pressure (PAP) [8].

Today, most bioimpedance measurement products are still based on electrodes connected in a star configuration to a central unit (e.g., a recorder or monitor), as shown in Figure 1. A current, typically a cosine wave $i_{2}=I\cos\omega t$ of magnitude I and angular frequency ω, is produced by a current source and injected into the body via one current electrode and drained by another connected to an opposing current source $i_{1}=-i_{2}$. Depending on their implementation, the two current sources may not match exactly. In this case, the mismatch current is drained through a third current electrode commonly called the neutral (N) or right-leg (RL) electrode (the left-most electrode in Figure 1). The RL electrode is also used to drain interfering currents picked up from the environment (e.g., 50 Hz from the mains), avoiding any risk of saturation of the electronics, in a mechanism known in biopotential measurement as the right leg electrode [9]. In Figure 1, the right leg electrode mechanism is shown with a functional diagram (in blue), highlighting that the voltage $v_{1}$ (electrode potential with respect to common ground) is measured at electrode #1 and subtracted from the setpoint (here 0). The error is filtered by the controller G (e.g., an integrator), and the resulting signal $u^{\prime }$ defines the voltage of the controlled voltage source. This control loop is usually implemented in analogue form with an operational amplifier (which performs the subtraction, the filter G, and the controlled voltage source $u^{\prime }$), but the use of a functional diagram allows for a more abstract description, which is also valid for a digital implementation, if this would be more favorable. The injected/drained current results in a voltage drop across the impedance to be measured Z, which is acquired in the same way as the biopotentials [9]. I/Q demodulation of this voltage (i.e., multiplication by $\cos\left(\omega t\right)/I$ and $\sin\left(\omega t\right)/I$ followed by low-pass filters) yields the ‘real’ and ‘imaginary’ parts of the impedance at a given angular frequency, i.e., $Z_{\omega }\left(t\right)=R_{\omega }\left(t\right)+jX_{\omega }\left(t\right)$, where $R_{\omega }$ is resistance and $X_{\omega }$ is reactance. By separating the electrodes for potentials (called potential electrodes) from those used for currents (called current electrodes), the impedance Z can be measured without being affected by the skin/electrode impedances (tetrapolar method [9]). The cables connecting the potential electrodes use a shield driven by a unity-gain follower amplifier at the measured potential to avoid decreasing the input impedance of the measurement amplifiers. Current electrodes also use a driven shield to prevent capacitive current leakage. For wearables using dry electrodes and/or high frequencies for current injection, the cables will leak a current into the body through capacitive coupling. To avoid this capacitive leakage current that could interfere with the measurement of Z, a second grounded shield can be added [2].

The implementation of bioimpedance sensors in medical garments is difficult for several fundamental reasons. One is the preferential use of dry electrodes, which makes the demand for appropriate shielding more pressing as higher voltages are required to inject a current through the higher skin/electrode impedance. Shielded and double-shielded cables are difficult to integrate into wearables, especially if many bioimpedance channels are to be measured. Using active electrodes [2,3] is a well-known solution to avoid shields, but they need to be powered. Powering them from the central unit is not trivial when considering medical safety standards. The basic safety standard [10] for medical electrical equipment (capitalized words are defined terms in the standard) requires that the patient leakage current be no more than 10 μA d.c. Additionally, the standards require that two means of patient protection (MOPP) are used for safety. Double insulation for conductive tracks can be considered as one MOPP, but the presence of body fluids like sweat or urine make this difficult to implement. For solid insulation to be considered as MOPP in a system with a working voltage of less than 60 V, it must pass a dielectric strength test at 500 V rms for 1 min under the worst expected conditions, for example, in the presence of body fluids and at the end of the expected service lifetime. Based on these challenges, alternative means of patient protection, such as electronic detection and the active limitation of tiny patient leakage currents, are desirable.

Another factor complicating the development of wearable bioimpedance sensors is the challenge posed by the conventional star configuration for electronics cabling. Using one cable per sensor requires routing many cables that often require shielding or power signals, leading to bulky devices. Additionally, each of the cables must be reliably connected to the central unit. Medical electronics that must be protected from moisture and body fluids necessitate the use of expensive and often large connectors.

To address these problems, a parallel bus architecture for sensors is appealing because it is virtually independent of the number of sensors [4]. Implementing the bus with as few wires as possible is ideal; however, doing so increases the complexity because several functions must occur simultaneously on the bus (command/data transfer, synchronization, potential/current reference, and powering). Such a system has been demonstrated with cooperative sensors [11,12,13,14], defined as active electrodes connected to a parallel bus with up to two wires, that addressed all of these challenges except for supplying power on the bus. Other works have demonstrated two-wire buses with communication and a power supply but without potential reference and current return for biopotential or bioimpedance measurements [15,16,17,18]. Recently, however, cooperative sensors measuring biopotentials have been extended to the power supply function, solving the problem of having a strong signal (power supply) that is orders of magnitude higher than the signal to measure [19]. Remotely powered cooperative sensors that measure bioimpedances have many similarities to those that measure biopotentials. However, certain key differences and complexities exist that are presented in this paper, which can therefore be considered as a companion paper to [19].

This paper is organized as follows: In Section 2, we present the prior state of the art in cooperative sensors, which used a dedicated battery per sensor, as well as a bootstrap circuit to achieve high input impedance and efficient current sources. This section highlights the functions that must remain unaltered by the introduction of power distribution in the same wires. In Section 3, we present two power distribution solutions: one called the ‘legacy approach with 500 Hz powering and off-the-shelf components’, which builds on the prior work outlined in Section 2, and the other named the ‘approach addressing the safety issue with powering at 1 MHz and ASIC’. The implementation details and experimental results are provided for both approaches in Section 4, while Section 5 reports the conclusions.

## 2. State of the Art in Cooperative Sensors

Cooperative Sensors refers to an electronics architecture wherein active electrodes are connected by up to two wires in a parallel bus. The sensors are ‘cooperative’ because the sensors must operate in a simultaneous, coordinated manner to obtain a difference in potential or to inject/drain a current. Compared to conventional approaches, cooperative sensors [19,20,21,22] benefit from a parallel bus arrangement, and unlike multi-wire multiplexing architectures [4], the complexity of their connection is reduced to only two wires for all functions.

### 2.1. Basic Circuits and Their Interconnections

Figure 2 shows a patented generic mechanism [11,12] for synchronization and control of the cooperative sensors (middle and right) and for transferring the acquired signal to the central unit (left). The latter includes the right leg (RL) electrode. For a given current channel, cooperative sensors can be potential or current electrodes. They are ‘potential electrodes’ when their purpose is to measure the potential (therefore with their current source in red in Figure 2 absent or set at zero current), and ‘current electrodes’ when their function is to inject a current $i_{1}$ (different from zero). The central unit uses a voltage source U to transmit synchronization information and other commands simultaneously to all cooperative sensors. The cooperative sensors receive the signal as a voltage across their current source in black, i.e., the voltage between the two bus wires. This received voltage then feeds the clock recovery and the synchronization block, S. The cooperative sensors communicate data such as the measured potential $v_{i}$ back to the central unit by modulating the current with the block M. For bioimpedances, the modulator M in Figure 2 implicitly includes an IQ demodulation to extract Z(t) before modulation for transmission. Note that the voltage $v_{i}$ in current electrodes can be used to measure the skin/electrode impedance. As this voltage is much higher than that of potential electrodes, the amplifier shall have a lower gain when $i_{i}\neq 0$. The composite information from all sensors is received and demodulated at the central unit by measuring the current in the bus. Signals from an individual sensor are then recovered by demodulation, D. Note that multiple modulation schemes are possible, including amplitude modulation or digital phase shift keying with the addition of a digitization step. The power supply of each cooperative sensor, for example, a battery, is abstracted by a grey box.

A benefit of the cooperative sensor architecture is that the two-wire bus can also be implemented using conductive textiles, as shown in Figure 2. In this case, one side of the cooperative sensor is used as a dry electrode, while the other side interfaces to the conductive fabric. Contact with the top conductive layer is then provided by the fastener. The form factor of the cooperative sensors can be made very small (e.g., 4 × 4 mm^2^) if implemented as an ASIC, while the dry electrode and the contacts to the fabric can be larger and flexible. Using this approach, the assembly of cooperative sensor textiles is virtually seamless and cable free, reducing the impact of wearability constraints such as flexibility, breathability, stretchability, and washability. The controller G of the central unit maintains approximately 0 V between the lower conductor and the body. 

The upper portion of the bus is not directly in contact with the patient but can still be touched, for example, by the patient’s hand. This situation presents a safety problem that can be solved by electronically detecting leakage currents and stopping the operation of the device if a threshold is exceeded. Intermittent system operation is undesirable though, so an additional layer of fabric could be used to insulate the top conductive side for most situations, while electronic detection could then be used in rarer exceptional situations. This approach could be extended by making the outer side of the additional fabric layer conductive and connecting it to the skin. This is similar to how class I ME equipment is designed [10] and provides one mean of patient protection. Furthermore, this extra conductive layer would shield the middle conductive layer with respect to EMC emission, providing greater immunity in demanding applications or environments.

Figure 3 shows two possible configurations for EIT, i.e., for the measurement of multi-channel bioimpedances. The orange area with the central impedance Z symbolizes the tissues and the bioimpedances to be measured. Figure 3a has two types of sensors, one with a potential electrode and one with a current electrode (with odd and even numbers, respectively), whereas the sensors in Figure 3b are all identical with either a potential or current electrode depending on their current being equal to 0 or not. Between electrodes ⑥ and ⑦, the ‘pass-through’ symbol indicates that there is virtually a direct connection (no skin/electrode impedance) between the back wire and the body tissues. This effect [21] is achieved by the RL electrode and its controller G (see Figure 2) that ensures that the potentials of the wire and the body tissue are identical regardless of the current flowing through the skin/electrode impedance at the RL electrode. The scheme in Figure 3a is generally preferred when all current channels are activated simultaneously (for instance, each with a carrier at a dedicated frequency). Figure 3b is more common, as the current channels have time carriers (i.e., time multiplexing). As a result, the current is zero most of the time, which means that at any given time, there is only one pair of sensors with current electrodes—all other sensors have a potential electrode. The current channels are usually an injection of a current by one current electrode (e.g., ②) and the draining of this current by another (e.g., ⑧). However, especially in the case of Figure 3a, more complex patterns can be defined. For instance, instead of having only one injection electrode, half of the current can be injected by sensor ② and a quarter each by sensors ④ and ⑯.

### 2.2. Floating Supply, Bootstrapping, Current Source, and Current/Potential Wire Separation

A more detailed view of the cooperative sensor design for measuring bioimpedances is shown in Figure 4 [13,14]. In this implementation, each cooperative sensor is separately powered by a battery which is cumbersome but intrinsically safe and simpler than remotely powered sensors [19]. The input impedance of the amplifier of $v_{i}$ is also the output impedance of the current source $i_{i}$, which is significantly increased by using power supply bootstrapping [15]. The increase in impedance allows for the easy implementation of an efficient current source, as illustrated in red in Figure 4 with a voltage source in series with a high resistance R. In this example, the impedance seen from outside is not R but $\left(1+g\right)R$, where g is the open-loop gain of the operational amplifier. Moreover, this simple current source has a rail-to-rail voltage range (in contrast with other current source implementations, e.g., Howland current source). Note that the electrode and its connection to the operational amplifier are shielded using the half potential of the power supply.

The current injected through the electrode returns via the body, the central unit (or another sensor with opposite current injection), the upper wire, and finally, the output of the bootstrap operational amplifier. The sensor measures the electrode potential with respect to the lower wire. As no (bioimpedance) current flows in this wire, the impedance of the wires does not affect the bioimpedance measurement. This feature is particularly important if the wires are implemented as conductive tracks in a garment, since their impedance may significantly vary when they are stretched. 

## 3. Methods

In this section, we present two different solutions for remote powering cooperative sensors used to measure bioimpedances. The approach is similar to that described in [19] for biopotentials but addresses the additional demands required for the measurement of bioimpedances. 

The first solution (Section 3.1) describes how remote powering is added to the two-wire bus without adding noise to the measure bioimpedances and without disrupting synchronization and communication between the sensors and the central unit. Section 3.2 then addresses how remote powering can be made safe for medical regulations while significantly reducing the volume of the sensors (e.g., from 7.5 cm^3^ to 0.3 cm^3^) and the power consumption (e.g., from 5.8 mA to 150 µA), while additionally detailing circuit strategies for the measurement, current injection, and management of return currents.

### 3.1. Previous Approach Using Off-the-Shelf Components and Powering at 500 Hz

A modified version of the circuit in Figure 2 is shown in Figure 5, which incorporates circuitry for remote powering. In this scheme, the voltage source U of the central unit transmits a powering signal over the bus from which the cooperative sensors can generate a DC supply. Harvesting power from the bus in this way is depicted as a current source whose current is in phase with the voltage U (power is consumed when current and voltage are in phase).

It is important that the power supplied from the central unit does not disturb the measurements. Although the resistance of the bus wires may be low, the supply current is high, so the voltage drop on the lower wire will add to the voltage of interest, e. To avoid interference, the power supply frequency can be placed outside the frequency band of the measurements. The first-order bandwidth of the bioimpedance is, for example, 40 ± 0.5 kHz for 25 current channels and 40 frames per second. Therefore, powering could be carried out at 0 Hz. However, if the biopotentials are to be measured simultaneously by the same sensors, [1] has shown that 0 Hz is not a good choice. The first solution described in [19] proposes powering with a square wave at 500 Hz. Interference with the bioimpedance measurement must be expected at 500 Hz and its harmonics, i.e., 1 kHz, 1.5 kHz, 2 kHz, etc. If the frequency band for biopotential measurements is chosen at, for instance, 40 ± 0.5 kHz, possible interferences could be expected at 39.5, 40.0, and 40.5 kHz. IQ demodulation will shift this band to −500, 0, and 500 Hz. The other harmonics of the powering will also be shifted but will remain at frequencies of 1 kHz, 1.5 kHz, 2 kHz, etc. When sampled at 1 kHz, all of these spectral lines will create aliases at 0 and 500 Hz, and therefore, an anti-aliasing filter must be used to reduce their power. A delta sigma analog-to-digital converter, as used for biopotentials in [19], can efficiently avoid aliasing at 0 Hz, but there will still be some aliasing at 500 Hz. Moreover, at 0 Hz, there will still be the original 40 kHz harmonic, which is small because it is the 100th harmonic. Moreover, for most applications, the useful bioimpedance signal is in the respiration or cardiac bands that do not include 0 Hz, i.e., a high pass comb filter is used to remove the 0 Hz component of the bioimpedance signal (which is for 25 current channels at 0 Hz, 40 Hz, 80 Hz, etc.). The 500 Hz alias can be removed with a notch comb filter at 20 Hz (i.e., the Nyquist frequency of the bioimpedance channels) if the number of current channels is odd (e.g., N=25) since 500 Hz is in the middle of the N bioimpedance spectra.

The harmonics generated by the 500 Hz power supply frequency are of low enough amplitude to avoid significant interference with the digital communication (the 2 Mb/s of [13,14] were modified to 1.28 Mb/s, in both directions, to conform to the frequency required by the delta-sigma converter to sample at 1 kHz). It was observed, however, that bits near the rising/falling edges of the 500 Hz power signal were disturbed, so these bits were removed from the communication payload. In our prototype, 110 bits were removed in total, reducing the throughput by 17%.

In Figure 4 and in [13,14], upstream communication (from the central unit to the sensors) is implemented as a voltage source and the downstream channel (from the sensors to the central unit) as current sources. Alternatively, the upstream and downstream channels could be interleaved through time domain multiplexing to allow both channels to use either voltages or currents to communicate. In Figure 5, the current sources are implemented with the Thevenin equivalent voltage source with a resistance in series because practically, this is easier to implement using the digital output pin of a microcontroller or FPGA. The capacitor, inductor, and resistor are chosen to create a first-order bandpass filter that filters the communication band from the biopotentials while simultaneously attenuating the high-frequency harmonics from the digital signal. The received signal is then measured as the voltage across the RLC circuit. The digital signal can then be reconstructed with a high-pass filter and Schmitt trigger before demodulation in the D block to obtain $v_{i}^{\prime }$.

Placing the cooperative sensors in parallel on the 2-wire bus practically limits the number of sensors that can be added. The combination of the LC circuit and the RLC circuit effectively acts as a voltage divider. The received voltage is thus the transmitter voltage divided by the number of receiving units (sensors and the central unit), which, in practice, limits the maximum number of sensors to approximately 25.

The powering signal is a 500 Hz square wave generated in the central unit (voltage source U). Since the inductances of the central unit and sensor are tuned for the higher communication frequencies, their presence has minimal impact on the supply current. The sensors can thus harvest energy from the 500 Hz signal using rectifier diodes to generate a positive and negative voltage rail across the storage capacitors.

In the implementation of Figure 5, there is no floating power supply and therefore no bootstrap as in Figure 4. The current source $i_{i}$ must be designed with high output impedance. Moreover, its current returns via both upper and lower wires, thereby affecting the measurement of the electrode potential since the lower wire is used as reference potential.

### 3.2. The Approach Addressing the Safety Issue with Powering at 1 MHz and ASIC

While the circuit in Figure 5 reduces power supply interference with bioimpedance measurements and with communication on the 2-wire bus, it does not address safety for medical devices. In medical device standards, the allowed patient leakage current at 500 Hz is 100 μA for type bf devices [10], which is already ten times higher than the allowed current at d.c. However, measuring a 100 μA leakage current by monitoring the power supply signal is a difficult task. Assuming a per sensor consumption of 8 mA (see next section) and 25 sensors, the supply current on the bus is 200 mA, so measurements must have an accuracy of 1 part in 2000. All sensors would additionally require a controllable current source buffer to ensure that each sensor consumes precisely 8 mA.

To better ensure compliance with the allowed patient leakage current, the power consumption of the sensors should be reduced (e.g., by a factor of 20 down to 400 μA), and the power supply frequency should be increased, e.g., to 1 MHz where the standards allow up to 10 mA of leakage current. These changes facilitate the detection of excessive leakage current (1 part in 1). It is important to note that the current measured in the bus is different from the current consumed by a sensor. The sensor itself consumes half of the 8 mA and 400 μA noted above, i.e., 4 mA and 200 μA, respectively. The difference is due to the conservation of energy: the supply voltage U is ±VCC/2 with a current of ±2I (rms value 2I for square waves), which allows sensors with the dual half-wave rectifier to have a VCC supply and I current for the electronics.

In our effort to reduce consumption, we developed an application-specific integrated circuit (ASIC) which further optimized the electronic functions detailed in Section 2. The ASIC implementation reduced power consumption by a factor of 20 and also eliminated the need to perform digitization at the sensors (analog-to-digital converter). The transmission of analog values instead of bits also increased the throughput and enabled the detection of other signals, such as sound picked up by a stethoscope embedded in the electrode. The 1 MHz power supply is now interleaved with the communication, i.e., in every two periods, there is one for power supply and one for communication [22]. Figure 6 depicts the design. The inductances are no longer needed, which is beneficial given the challenges of integrating inductors in silicon. The rectifier diodes allow the storage capacitors to be recharged (harvesting period) when the voltage U on the bus is high enough. When the voltage U is low enough for the diodes to be in a blocked state, the communication current source of one sensor is used to transmit the information to the central unit. For the other sensors, the current source is disabled (i.e., no current is consumed by these sensors during the communication period). Therefore, the central unit decides if the period is a harvesting or communication period with the level of the voltage U. As shown in Figure 7, by choosing whether the harvesting period follows or precedes the communication period, the central unit can send a 0 or 1 bit to the sensors (Manchester code). This upstream communication channel may be used, for instance, to configure the sensors. Note that the Manchester code always has a transition (H to L or L to H) in the middle of its period (see blue edges in Figure 7). Therefore, two harvesting periods in a row (H to H separated by a blue line) can easily be interpreted by the sensors as a synchronization marker rather than a 0 or a 1. From this marker, every sensor can count the number of edges and recognize its communication period defined by its ID. Note that this scheme is similar to that described in [19] but simpler in the sense that the edges are always regular and can readily be used as clock signals (without the need for PLL or timer). 

The regulated supply rails, VCCF and GNDF (specific to each sensor), can be bootstrapped using a follower to manage the reference potential of the LDO voltage regulators. Assuming an LDO gain, g (where the LDO produces a current i=gu, with u representing the voltage error at the LDO output), the input impedance of the open-loop circuit is increased by a factor of gz at low frequencies. To achieve high gain at low frequencies, $z=z_{a}+z_{b}$ should behave like a capacitance (see Figure 6). For stability at higher frequencies, however, it is preferable for *z* to resemble a resistance. The open-loop input impedance is essentially determined by the follower’s input impedance (typically 10 pF). Bootstrapping magnifies this impedance by gz, allowing the circuit to reach very high input impedance at low frequencies [22]. Compared to Figure 5, which lacks bootstrapping, this approach also improves the efficiency and natural shielding of the sensor input, as the ground and power rail planes inherently provide driven shielding. Additionally, the output impedance of the current source (R in Figure 5) as seen from the body is magnified by the bootstrap. If z is an order of magnitude larger than the lower wire impedance (which is easily the case), the part of injected current $i_{i}$ that is conducted by the lower wire is negligible. As a result, the bioimpedance measurement is not affected by the wire impedance.

Before transmission, the biopotential is filtered by the voltage dividers $z_{a}$ and $z_{b}$, amplified, and IQ demodulated. The filter is a band-pass filter centered at 50 kHz (EIT current frequency of the ASIC variant) that removes the electrode offset (up to 300 mV) and possible biopotentials (generally lower than 10 mV). A perfect IQ demodulation would not require such filtering since the multiplication of the signal by a cosine at 50 kHz would swap the bioimpedance and biopotential bands. But in practice, it cannot be perfect, and the high-pass filter will prevent residual biopotential in the demodulated bioimpedance signal. Most of the bioimpedance signal is at or close to 0 Hz (typically 98–99%). However, the signal of interest is generally at breathing or cardiac frequencies. Therefore, compression of the low frequencies (with a filter inverse to a low-pass filter) is advisable before transmission. At reception by the central unit, the original signal can be decompressed with a low-pass filter. When the bioimpedance signal is the result of more than one current channel, such compression filter has to also be applied and shifted at 40 Hz, 80 Hz, etc. (comb filter). This pre-processing step improves the signal-to-noise ratio of the analog communication. Figure 8 shows a possible implementation of the comb filter with transfer function $1-\alpha z^{-N}$ with $z=e^{jf/f_{s}}$ (z transform variable), $f_{s}=1$ kHz (sample rate), N=25 (number of current channels), and
$$\alpha =\frac{1}{1+a\left|1-e^{-j2\pi Nf_{c}/f_{s}}\right|}=0.996$$
where $f_{c}=0.25$ Hz (corner frequency chosen here as the lowest respiration frequency) and a=0.1 (compression factor of the bioimpedance signal at 0 Hz). The current sample voltage is applied over $C_{r}$ and $C_{s}$ in series, with the capacitance $C_{r}$ having been shorted (switch r closed) just before. The charge accumulated by $C_{r}$ and $C_{s}$ in series is copied, thanks to the operational amplifier, in the capacitance $C_{t}$ (also shorted just before). The resulting transfer function is therefore
$$-\frac{1}{C_{t}}\frac{{C_{r}C}_{s}}{C_{r}+C_{s}}\left(1-\frac{C_{r}}{C_{r}+C_{s}}z^{-25}\right)$$

When the switch s is open, the charges in the capacitance $C_{t}$ are held (sample and hold function), and the potential $v_{o}$ is ready for transmission. While the switch s is open and as soon as the voltage $v_{o}$ has been transmitted, switches r and t can be closed. They are reopened before another switch s is closed for the next sample. The memorized value in $C_{s}$ is available 25 samples later.

The M modulator (see Figure 6) selects the correct time slot for the sensor to communicate. All sensors are synchronized to sample their electrode potential simultaneously, and the measured value to be transmitted is stored in a capacitor until being transmitted. While this paper describes the measurement of bioimpedance, the developed ASIC can also measure biopotentials [1] and, if interfaced with an electret, body sounds (stethoscope).

### 3.3. Comparison to Existing Work

A comparison with existing work is presented in Table 1 to highlight the novelty of this work. Remotely powered cooperative sensors are unique in that they do not require a local power supply (e.g., one battery per sensor), enabling miniaturization and easier integration, among other benefits. Compared to the work presented in [1] where only the measurement of biopotentials was addressed, this paper focuses on the measurement of bioimpedances and the needed additional circuits. Note that being able to monitor the leakage currents allows wearables to be compliant with medical safety standards without relying on the insulation of conductors in a garment and without the need for waterproof connectors.

## 4. Results

### 4.1. Legacy Approach with 500 Hz Powering and Off-the-Shelf Components

#### 4.1.1. WELMO Vest

Figure 9 shows the WELMO vest [23] developed in the context of a European project for the remote detection of patients with COPD exacerbation of the lungs [23,24,25,26,27,28,29,30,31,32,33,34,35,36]. The EIT chest strap is based on the approach shown in Figure 5. The central unit is the small device visible at the bottom of the vest. The central unit is powered by an IEC 62 133 battery and is used to record data and provide wireless communication. The sensors are designed around a complex programmable logic device (CPLD) that implements the PLL and internal clock reconstruction, interfaces to a 24-bit delta-sigma converter, and communicates to the central unit over the bus. The sensors interface to the two-wire bus through two metal electrodes in the top part of their plastic housings. The wires in the belt end on attachment washers (realized as PCBs made of FR4) onto which finger spring contacts are mounted. The sensors are brought into contact with the finger springs by screwing them onto the attachment washers. For this, there are threaded holes in the electrodes in the top part of the plastic housings (see the photo in the bottom line in the middle row in Figure 9). The housings further protect the electronics against moisture (IPX6 according to IEC 60 529).

A simplified electronic schematic implementing the principles from Figure 5 is shown in Figure 10. The central unit is connected via a two-wire bus to a chain of cooperative sensors. The voltage source U of the central unit (see Figure 5) is implemented with switching transistors that alternate between the battery (4.6 V) and a bypass (0 V). Both the central unit and the cooperative sensors harvest power from the two-wire bus with dual half-wave rectifiers. The offset of U (2.3 V) is removed by a capacitor connected upstream from the half-wave rectifiers. The capacitor provides a bipolar VCC/GND supply (±2.3 V) that is symmetrical with respect to REF, the potential of the lower line of the two-wire bus. To communicate over the two-wire bus, an impedance is connected to a digital output of the CPLD (Logic), creating a Thevenin equivalent to a current source. To receive data, a Schmitt trigger is used to regenerate the edges of the received signal and output them to the CPLD.

To keep implementation simple, the central unit in Figure 10 has two electrodes (textile electrodes ① in Figure 9). The same approach for communication is used as in [14]. Two resistances define the gain of the first amplification stage ②, with the sensor electrode being connected to the operational amplifier’s positive input. The signal is then IQ demodulated thanks to multiplication with a square wave implemented with two double switches ③ and an RC low-pass filter ④ (in Figure 10, only the resistive part of the bioimpedance is demodulated). This RC low-pass filter implements the coarse antialiasing filter required by the delta-sigma converter (i.e., a more aggressive filtering is obtained by the digital filter of the delta-sigma converter). An instrumentation amplifier ⑤ allows for a last stage of amplification before the delta-sigma converter.

The current source for the injection of the current is implemented by the circuit in red (see Figure 10). The CPLD produces the square wave digital signal according to the pattern of the current channels. An instrumentation amplifier allows for this square wave voltage to be applied across the resistance for its translation to a current. The capacitance prevents any current at low frequencies to disturb biopotential measurements (not shown in Figure 10, see [19]).

The WELMO vest has 18 cooperative sensors, 6 sensors for measuring chest sounds (stethoscopes), and 16 for EIT (some sensors combine both functions). The current channels are defined as shown in Table 2. Note that with this definition, the 16th channel is linearly dependent on the others, and therefore, with this definition, the information is approximatively equivalent to 15 current channels.

The average noise when measured on a 56 Ω resistance is 45 mΩ (rms) in the full bandwidth (0–20 Hz). However, a few sensors have a noise of 15 mΩ (rms), and one exceeds 100 mΩ (rms). The reason for this dispersion could not be identified yet. In the respiration band (0.25–2.5 Hz), the average noise is reduced to 15 mΩ (rms). By increasing the injection current by a factor of 10 in the next development, we expect to reduce this noise to about 2 mΩ, a target similar to [7]. The new current will then be at 1 mA, still compliant with the safety standard [10] that allows a maximum at 4 mA for 40 kHz.

#### 4.1.2. Calibration

The injected currents by the EIT device are denoted by the matrix i (16×25) and the acquired potentials by the matrix u (16×25), where the rows correspond to the sensors and the columns to the current channels. Due to the imprecision of the electronics, the injected currents i may slightly differ from the intended currents I and the acquired potentials u from the electrode potentials U. Performing EIT processing with $\left(I,u\right)$ instead of $\left(i,U\right)$ may result in errors since the resistance (or impedance matrix) R (16×16) maps i↦U, i.e.,
U=Ri

Therefore, it is better to calibrate the EIT device, i.e., find the function that maps $\left(I,u\right)\mapsto \left(i,U\right)$ so that $\left(i,U\right)$ can be used for EIT processing. The following model is considered:i=GIQ, u=HUP+v
where Q, P are 25×25 matrices, G, H are 16×16 matrices, and v is a 16×25 matrix. Matrices G and H are diagonal because there is no possible coupling between sensors (only the gain of the current sources or amplifiers must be modeled), and matrix Q can be arbitrarily set equal to the identity matrix (since Q and P appear in a product in U=Ri). If x= $\left(H,P,G,v\right)$ are known, then $\left(i,U\right)$ can be obtained from $\left(I,u\right)$ with
$$i=GI,\;U=H^{-1}\left(u-v\right)P^{-1}$$

The values of x= $\left(H,P,G,v\right)$ can be computed with a least mean square approach, i.e., by defining a quadratic function $f\left(x\right)$ that has its minimum at the solution. The function f is defined as
$$f=\sum_{i}{\left\|w^{\left(i\right)}\right\|}^{2}$$
where $w^{\left(i\right)}$ is a vector of dimension 1057 (16 + 25 × 25 + 16 + 16 × 25) stacking the elements of the following matrix:$$u^{\left(i\right)}-S\circ \left(HR^{\left(i\right)}GIP+v\right)$$
where $u^{\left(i\right)}$ and $R^{\left(i\right)}$ are the potential and resistance matrices corresponding to configuration $\left(i\right)$, ∘ is the element-wise multiplication (.* in Matlab language), and S is a matrix that has all its elements equal to 1 except for those equal to 1/g corresponding to current-injecting electrodes:$$S_{ij}=\left\{\begin{array}{c} 1\;\mathrm{if}\;I_{ij}=0\;\\ 1/g\;\mathrm{otherwise} \end{array}\right.$$
where 1/g is the attenuation gain of the amplifier measuring the potential of the current electrodes. 

A gradient descent method finds the minimum x of $f\left(x\right)$ starting with an initial guess $x_{0}=\left(1,1,1,0\right)$, where 1 and 0 are the identity and zero matrices. The iteration equation is
$$x_{k+1}=x_{k}-\left[f\left(x_{k}+\delta \right)-f\left(x_{k}-\delta \right)\right]$$
with δ being a small increment (e.g., for every entry, in turn).

Sixteen configurations $\left(i\right)$ were measured with two resistors r. The two first configurations are shown in Figure 11 (with the others being a rotation by one electrode further on the same model as from configurations $\left(1\right)$ and $\left(2\right)$ in Figure 11). 

Figure 12 shows an example corresponding to the configuration $\left(1\right)$ of the errors u−RI (top) and U−Ri, i.e., after calibration (bottom). The optimization process resulted in an improvement by an order of magnitude (from ≈ 5 Ω to ≈ 0.5 Ω). Overall, after calibration, the accuracy of WELMO when measuring a resistance of 56 Ω is 0.5 Ω, i.e., better than 1%.

### 4.2. Addressing Safety with Powering at 1 MHz and an ASIC Implementation

#### 4.2.1. ASIC Architecture

The ASIC architecture is similar to that described in [19], except for the injection of the current and potential amplification, AM demodulation, and comb filter. The ASIC has a power consumption of 0.67 mW (400 µA at 1 MHz), is manufactured in 180 nm technology, and has a footprint of 1.8 × 2.1 mm^2^. The architecture comprises two main sections (Figure 13). The first section, in red, implements the interface to the two-wire bus. Upon boot-up, the power management unit harvests energy and first enables internal power supplies. Once the internal supplies are stable, a delay-locked loop waits for a synchronization marker in the 1 MHz square wave. The synchronization marker is a break in the periodicity of the signal used to mark the start of one million periods, i.e., there is sync marker every 1 s [19]. Energy harvesting and communication occur on alternating periods of the power signal. Within the communication period, time-division multiplexing is used based on an ASIC’s unique ID so that only one sensor communicates its sample to the central unit at a time.

The second section of the ASIC, shown in green in Figure 13, provides the injection of the current and signal chains for the measurement of the electrode potential as results of the current injection of other sensors. This signal chain includes amplification, AM demodulation (multiplier) to shift the bioimpedance signal in baseband, and the comb filter. Moreover, additional sensing functionality, such as a stethoscope, presented in Figure 13 by an electret, an amplifier, and a filter, is also part of the ASIC.

The first part of the potential acquisition chain is a unity gain buffer. The output of the reference buffer is used as a floating reference in that it provides the ground reference for the positive and negative supply rails [19]. This bootstrapping approach enables the buffer’s power supply to follow changes in the potential of interest. Consequently, the voltage on the parasitic capacitances at the input is nearly 0 V, and virtually no current flows, effectively increasing the input impedance of the amplifier and the output impedance of the current source.

#### 4.2.2. The Architecture of the Central Unit

Figure 14 presents the central unit, which is identical to the description in [19]. In this implementation, the bus wires A and B are energized with a voltage source U providing a 1 MHz square wave of amplitude $\pm V_{h}.$ As mentioned in Section 4.2.1, powering and communication occur in alternating periods of the 1 MHz signal, with each sensor having a specific time slot in which to transmit its measurement. During the appropriate time slot, a sensor will inject a quantity of charge onto the bus that is proportional to the measured potential. An ADC is used to sample the voltage at the terminals of the capacitor, after which a microcontroller can process and demodulate the value. Following a sampling period, logic in the central unit resets the transimpedance capacitor via a switch. The controller G in Figure 6 is implemented with a pass-through, as shown in Figure 10.

#### 4.2.3. Sensor Housings

The sensor housings shall best reflect the small volume made possible by ASIC electronics, be simple to assemble, protect the electronics from mechanical shocks and moisture, and provide the electrical contacts with the two wires and the skin. 

Figure 15 shows a cut view drawing implementing these requirements. An ASIC is mounted on a PCB ①, together with few additional electronic components (such as spring contacts, shown in yellow in Figure 15 on the top and on the bottom of the PCB). The housing itself is made of a bottom piece ② and a top piece ④ which are glued together during assembly and thus provide a watertight enclosure. An electrode which contacts the skin ③ is over-molded in the bottom piece, and two electrodes ⑤ which contact the wires are contained in the top piece. Note that Figure 15 only shows one half of the sensor, and therefore, only one electrode contacting the wires is shown. A clamp ⑧ presses the sensor onto the wires, which are realized as stripes of electrically conductive textile ⑥, and mounted on the surface of a thin compressible 3D knit ⑦. The springiness of the 3D knit provides the contact force between the electrically conductive stripes and takes up possible dimensional tolerances in the assembly. The clamp is secured on the other face of the textile belt ⑩ with a semi-rigid reinforcement ring ⑨. While the textile conductive tracks and the 3D knit are continuous, the textile belt has holes into which the sensors are placed. The sensor is fixed on the clamp with a nose similar to the one on the reinforcement ring and a notch in the housing. 

#### 4.2.4. Implementation Results

The verification of the ASIC and central unit was carried out with the setup shown in Figure 16. Two sensors (ASIC) are configured to let a current (100 µA) circulate through a resistor and two others to measure the resulting voltage drop. The maximum range is −255 to 255 Ω (the value is negative when the resistance current is opposite to the resistance voltage). Therefore, 0.02% resistors of 0, 50, 100, 150, 200, and 250 Ω are used for the test. Table 3 provides the obtained results. The linearity is better than 0.3% FS. The noise is about 30 mΩ rms in the 0.25–2.5 Hz respiration band, i.e., similar to the noise obtained with the legacy approach of Section 4.1. A higher current than 100 µA can be used to reduce this noise toward the targeted 2 mΩ rms [7]. The frontend amplifier can also be improved (a misunderstanding of the requirement during the ASIC design is at the origin of this discrepancy, because to obtain 2 mΩ rms, the amplifier noise must not be at 2 mΩ × 100 µA but at this value divided by the number of channels, which is 25 here). Tests with more than four sensors will be possible after the correction of another bug in the ASIC (see also [19]).

The input impedance (*C*_in_) of the frontend amplifier was measured with the setup shown in Figure 17. The sensor is in the potential measurement mode (EIT current set to zero), while a voltage source at 50 mV/50 kHz with a series capacitance of 10 pF is connected to its input. The 10 pF capacitance makes a voltage divider with the input impedance. Therefore, measuring the voltage across the 10 pF capacitor allows for the estimation of the input impedance. Thanks to bootstrapping (see Section 3.2), the overall input capacitance is lower than 0.2 pF.

Figure 18 shows how the ASIC sensors are realized as thin, coin-sized parts integrated in a textile harness. As discussed in Figure 9, the harness contains textile electrodes and tightly fits the body to ensure good contact between the sensors and the skin. The sensors are mounted on a belt which is secured to the harness with loops. The belt contains two conductive tracks, realized as tapes glued to a woven stripe and placed on top of a 3D knit (see middle row of Figure 18). A custom-made clamp attaches the sensors to the belt and presses them against the conductive tapes. The assembly is finished by placing the belt textile containing a reinforcement ring on top of the conductive tapes. During this assembly step, the noses on the clamp pointing outwards in the middle row of Figure 18 snap in place on the reinforcement ring.

## 5. Conclusions

This paper demonstrates circuitry for the remote powering of cooperative sensors operating on a two-wire parallel bus in the context of multi-channel bioimpedance measurements, specifically electrical impedance tomography (EIT). This architecture significantly reduces the amount of cabling while adding essential safety features, enabling the future development of wearable textile devices for medical imaging. 

A second contribution of this paper is the development of high-frequency, remotely powered ASICs with the added ability to detect hazardous leakage current. This new variant benefits from the original bootstrapping approach that increases the input impedance of the frontend amplifier and the output impedance of the current source. In this paper, the demonstration was limited to two current injection sensors and two potential measurement sensors. Future work will address defects in the implementation to enable the addition of more sensors. 

In addition to demonstrating a solution for combining multiple functions simultaneously on a two-wire bus (e.g., remote power supply, microvolt potential measurements, synchronization, and communication) this paper addresses how such wearable sensors can be made compliant with medical safety standards. Although leakage current detection was not specifically implemented for this paper, the problem was addressed by analyzing the allowable currents at alternative powering frequencies at which the central unit could more easily monitor the current. 

## 6. Patents

The work presented in this paper is based on the patent in [22].

## Figures and Tables

**Figure 1 sensors-24-05896-f001:**
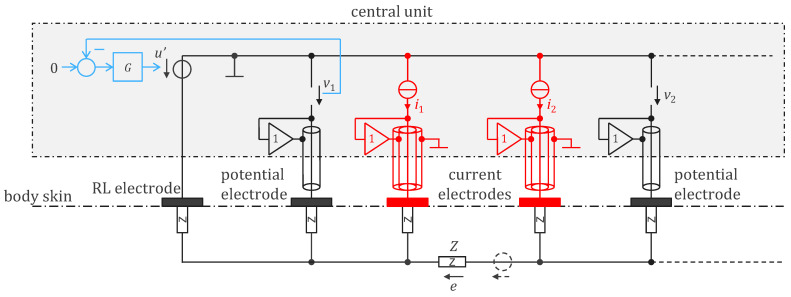
The conventional approach to measuring a bioimpedance $Z\left(t\right)$. Two current electrodes (in red) are connected with double-shielded cables to the central unit where a current source $i_{2}$ injects a current through the skin. The current flows through the impedance to be measured Z and is drained by another current electrode driven by the current source $i_{1}=-i_{2}.$ Any practical deviation between the two current sources flows through the RL electrode (called the right leg electrode because it was originally developed for ECG and placed on the right leg). The current $i_{2}$ is translated across the impedance Z by a voltage drop $e=Zi_{2}$ measured in the same way as biopotentials (difference $v_{2}-v_{1}).$ The controller G driving the voltage source $u^{\prime }$ allows the common ground potential to be set equal to the body potential, thus avoiding possible saturation of the electronics due to disturbing currents picked up in the environment and flowing through the skin/electrode impedance of the RL electrode. When the impedance is measured at a given angular frequency ω, it can be decomposed into a real part and imaginary part: $Z_{\omega }\left(t\right)=R_{\omega }\left(t\right)+jX_{\omega }\left(t\right)$. Furthermore, the current is a cosine wave $i_{2}=I\cos\omega t$, and the resistance $R_{\omega }\left(t\right)$ and reactance $X_{\omega }\left(t\right)$ are extracted from the voltage e with IQ demodulation, i.e., multiplication of the voltage e by $\cos\left(\omega t\right)/I$ and by $\sin\left(\omega t\right)/I$, respectively, followed by low-pass filters.

**Figure 2 sensors-24-05896-f002:**
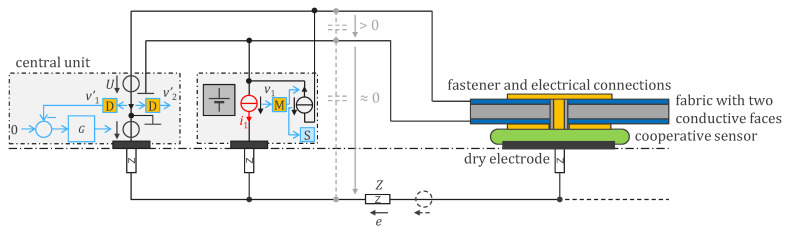
Cooperative sensors are active electrodes with additional circuitry that enables their connection via a parallel bus with up to two wires. The sensors communicate their measured data to a central unit, which also provides a synchronized clock. In applications not required to be defibrillator proof, the parallel bus can be made from conductive fabric. In this case, the controller G of the central unit maintains the voltage between the lower textile and the body at nearly 0 V, removing the need for bottom-side insulation. The top conductive textile can easily be insulated with an additional layer of fabric (e.g., a regular garment) if the excess leakage currents are electronically monitored. Highly integrated cooperative sensors can be attached and connected to the fabric, making the assembly seamless while maintaining the usual properties of the fabric (flexibility, stretchability, breathability, and washability). Cooperative sensors can be current electrodes (when the current $i_{1}$ is different from 0) or potential electrodes (when the current $i_{1}$ is zero). Symbol legend in Appendix A.

**Figure 3 sensors-24-05896-f003:**
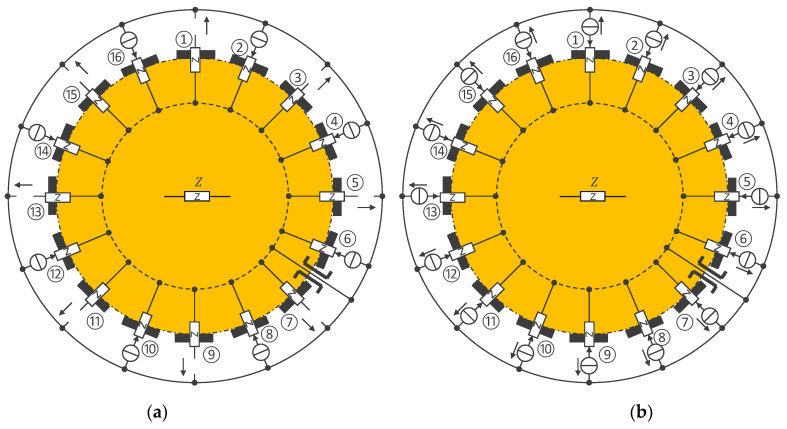
The connection of 16 sensors around a body part (e.g., chest or limb) for EIT measurements. (**a**) A device with two different types of sensors, one with a potential electrode and one with a current electrode. (**b**) A device with a single type of sensor with a potential or current electrode depending on the current/function (equal to 0 for potential electrode; different from 0 for current electrode). The symbol legend is in Appendix A.

**Figure 4 sensors-24-05896-f004:**
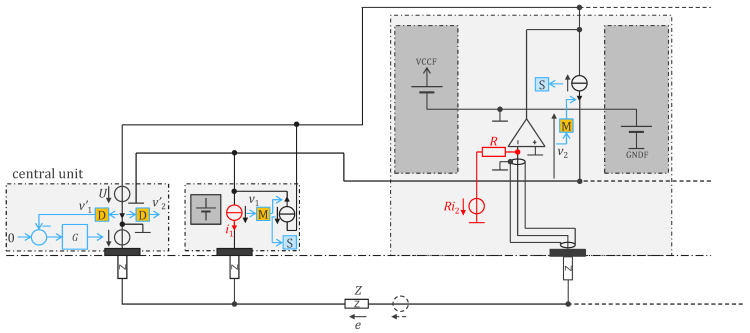
A simple bootstrap circuit is used to achieve extremely high impedance (input impedance for potential electrode and output impedance for current electrode) by leveraging the floating battery in each sensor. The parts added for the measurement of bioimpedances are shown in red—the other parts are the same as for biopotentials only [1]. The implementation of the current source (in red) can be simple thanks to the bootstrapping circuit [15] that significantly increases the open-loop impedance and has a rail-to-rail voltage range. The current return for the red current source comes from the upper wire only. As the lower wire is used for the measurement of potential, the impedance of the wires does not affect the measurement of bioimpedance. Patients are protected from leakage currents by diodes (not depicted) that prevent stored charge from leaving a sensor while simultaneously enabling the recharging of the batteries through the two-wire bus when the system is not being worn. See Appendix A for a symbol legend.

**Figure 5 sensors-24-05896-f005:**
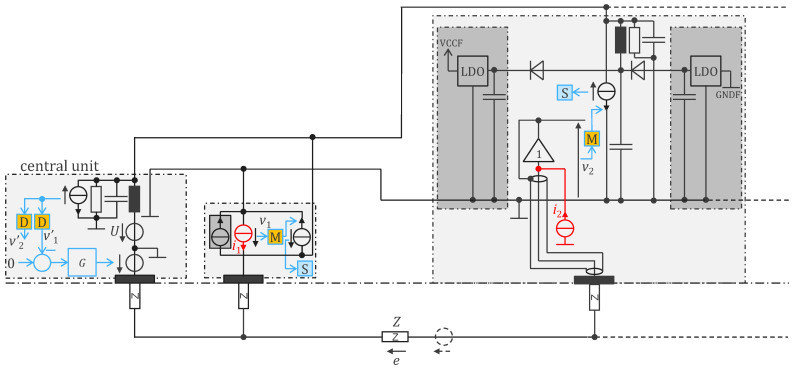
Remotely powered cooperative sensors for bioimpedance measurement with dry electrodes, with digital communication at 1.28 Mb/s in both directions (full duplex) and remote power supply at 500 Hz. Symbol legend in Appendix A.

**Figure 6 sensors-24-05896-f006:**
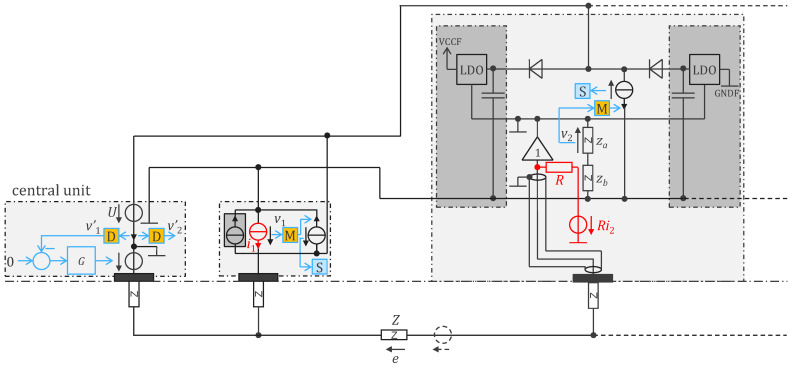
Remotely powered cooperative sensors for bioimpedance measurement with dry electrodes, with analog communication at 500,000 samples per second and remote supply voltage U at 1 MHz. Left: schematic overview of central unit circuit; middle: schematic overview of sensor circuit; right: detailed circuit diagram of sensor. Symbol legend is shown in Appendix A.

**Figure 7 sensors-24-05896-f007:**
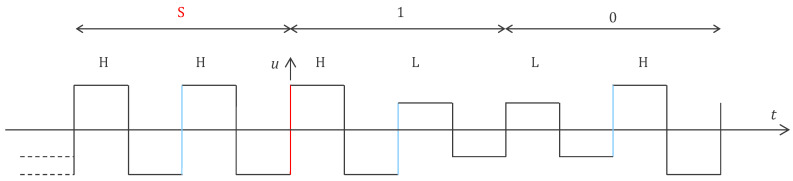
A supply voltage U consisting of a 1 MHz square wave with a sync marker (periodicity break) consisting of an HH period with a Manchester edge (in blue) every 1 s (every 1,000,000 periods of the 1 MHz square wave). The other periods contain a powering period H and a communication period L. If the period is HL, the sensors understand it as a 1, whereas LH is understood as 0. This upstream digital communication can be used to configure or control the sensors. The sensors harvest energy during subperiod H, and one of them (determined by the sensor ID and the position of the period with respect to the synch marker, shown in red in the figure) transmits an analog value during subperiod L.

**Figure 8 sensors-24-05896-f008:**
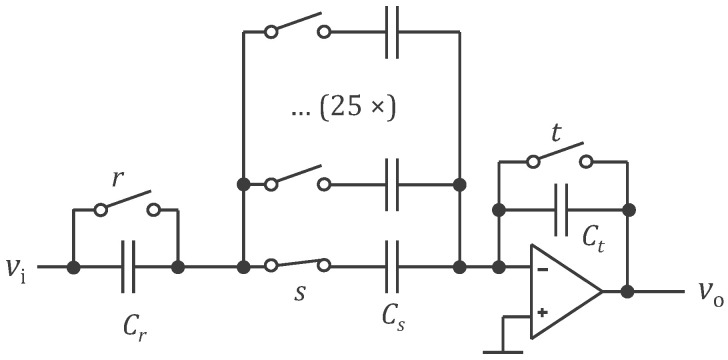
A possible implementation of the comb filter. The symbol legend is provided in Appendix A.

**Figure 9 sensors-24-05896-f009:**
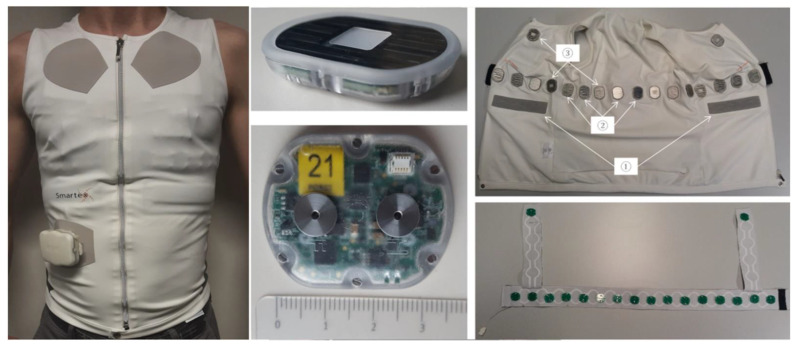
WELMO vest with embedded EIT chest strap with off-the-shelf components (textile part made by Smartex in framework of EU project WELMO). Left: worn vest, middle top: front view of cooperative sensor with stainless steel dry current/potential electrode and stethoscope (center), middle bottom: back view of cooperative sensor with its two connectors to 2-wire parallel bus, right top: open vest with embedded EIT chest strap with reference and RL textile electrodes ① and cooperative sensors with dry electrode ② and stethoscope ③, right bottom: back view of EIT chest strap showing 2-wire parallel bus and attachment washers.

**Figure 10 sensors-24-05896-f010:**
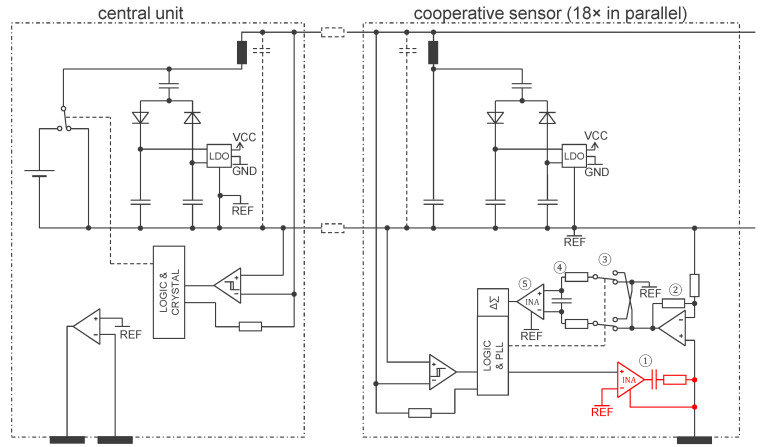
A possible implementation of the principles shown in Figure 5 (as prototyped in the device shown in Figure 9). Note that the safety protection circuit is not pictured for simplicity. The symbol legend is provided in Appendix A.

**Figure 11 sensors-24-05896-f011:**
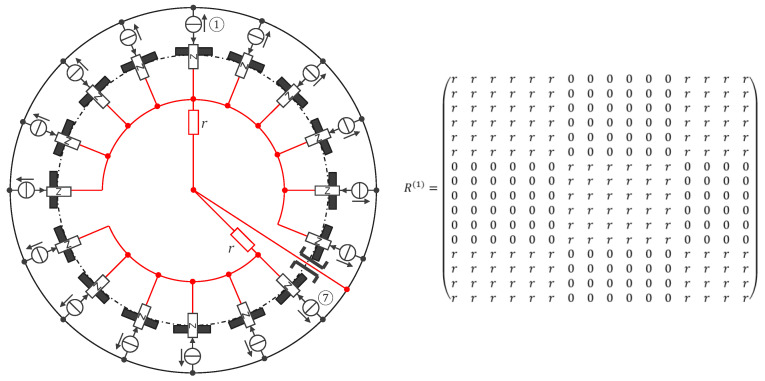
Configurations $\left(1\right)$ and $\left(2\right)$ ((**top**) and (**bottom**)), where the EIT device (left and in black) is connected to two resistors r (in red) to provide information corresponding to different resistance matrices $R^{\left(1\right)}$, $R^{\left(2\right)}$, etc. (right), for the optimization function f, allowing to compute by optimization the calibration function $\left(I,u\right)\mapsto \left(i,U\right)$. The symbol legend is shown in Appendix A.

**Figure 12 sensors-24-05896-f012:**
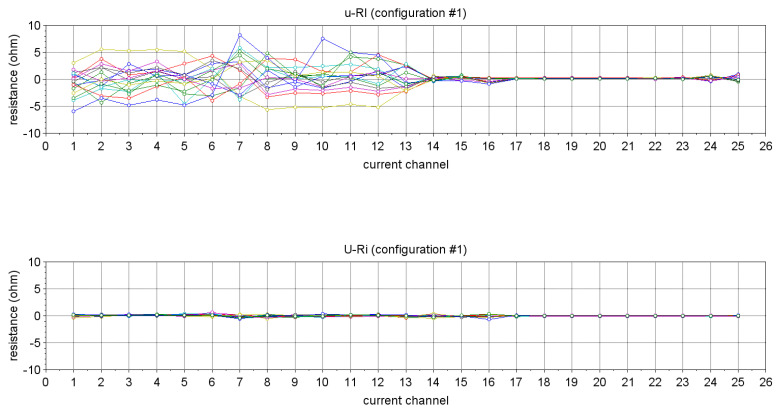
Errors u−RI (**top**) and U−Ri, i.e., after calibration (**bottom**) for configuration $\left(1\right)$. Comparable results are obtained for other configurations $\left(2\right)$ to $\left(16\right)$.

**Figure 13 sensors-24-05896-f013:**
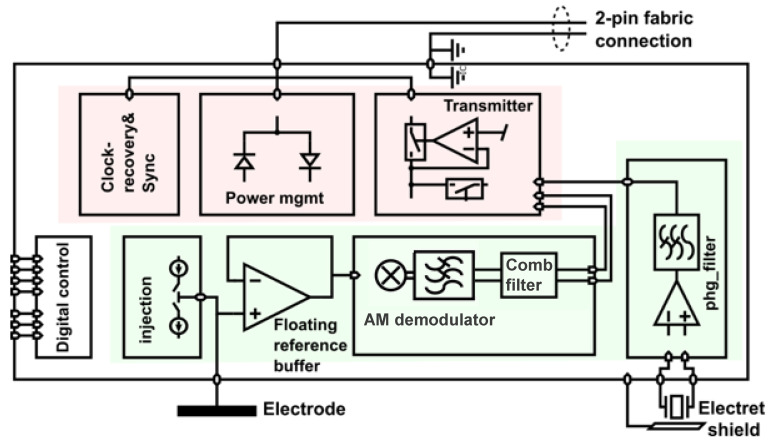
A block diagram of the ASIC implementation of cooperative sensors for bioimpedance measurements. The circuit blocks that interface with the 2-wire sensor bus are marked in red, and the signal processing circuits are in green. The symbol legend is shown in Appendix A.

**Figure 14 sensors-24-05896-f014:**
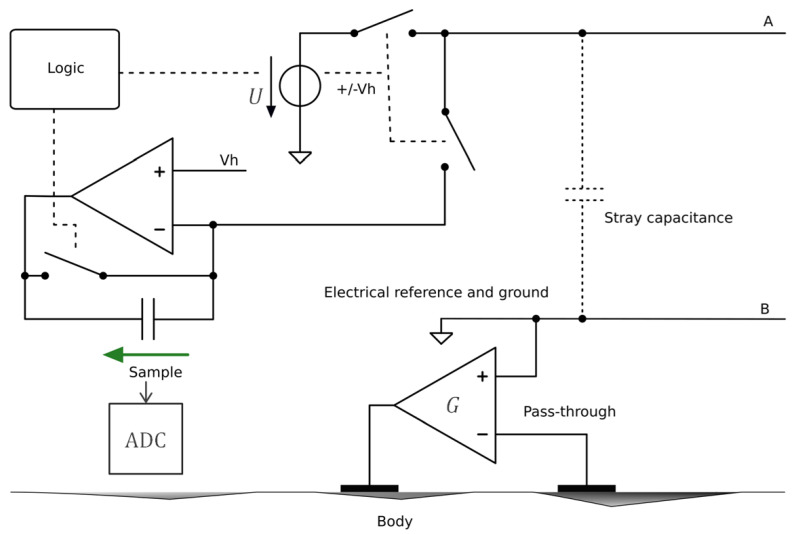
A diagram of the central unit based on the approach shown in Figure 6. The symbol legend is shown in Appendix A.

**Figure 15 sensors-24-05896-f015:**
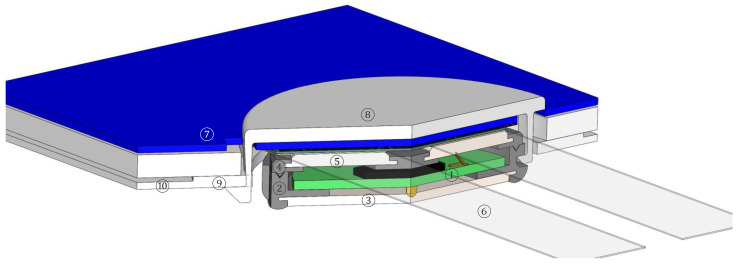
A cut view of the integration of a sensor realized with an ASIC. ① A PCB (green) with a mounted ASIC (black) and finger spring contacts (yellow). ② The bottom part of the housing with an over-molded stainless steel skin electrode ③, with the connection between the PCB and the electrode being obtained with a spring contact (in yellow). ④ The top part of the housing with over-molded wire contacts ⑤ (only one is shown). ⑥ An electrically conductive track on a slightly compressible textile ⑦. ⑧ A clamp pressing the sensor onto the textile. ⑨ A reinforcement ring on belt textile ⑩. The height of the ASIC sensor (without a clamp and without textile) is 4.7 mm. The size of the sensor can be reduced if only the bioimpedance is considered (in our development, we had sensors that also included a stethoscope, not shown in this figure).

**Figure 16 sensors-24-05896-f016:**
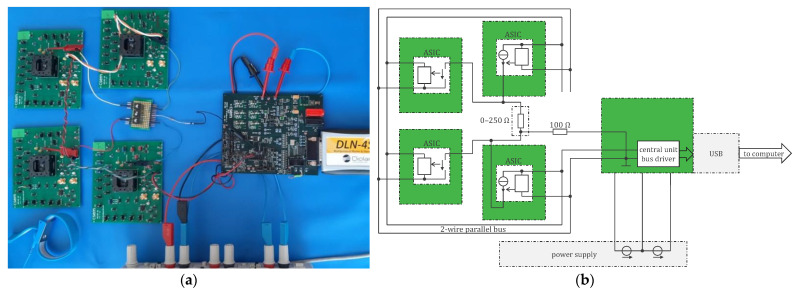
(**a**) Real setup and (**b**) functional diagram of setup for first verifications of concept including four sensors, i.e., ASIC (left), central unit (right), and resistance to measure (center).

**Figure 17 sensors-24-05896-f017:**
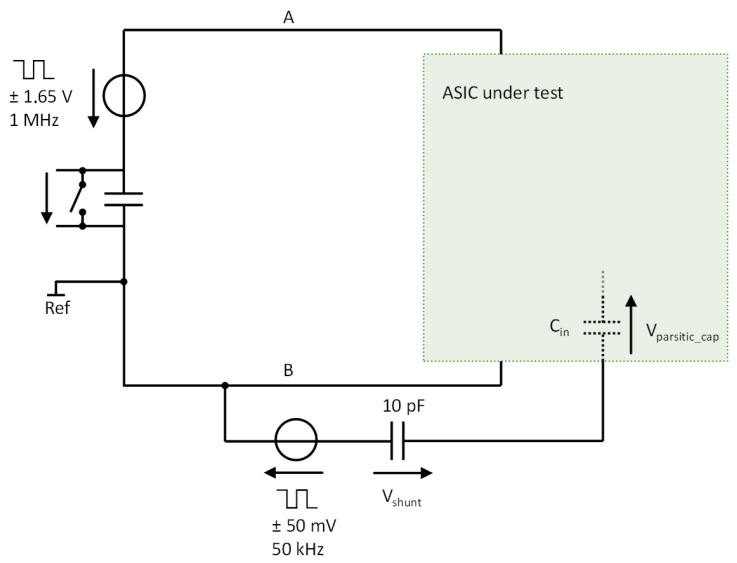
The setup used to measure the input impedance of the ASIC frontend amplifier.

**Figure 18 sensors-24-05896-f018:**
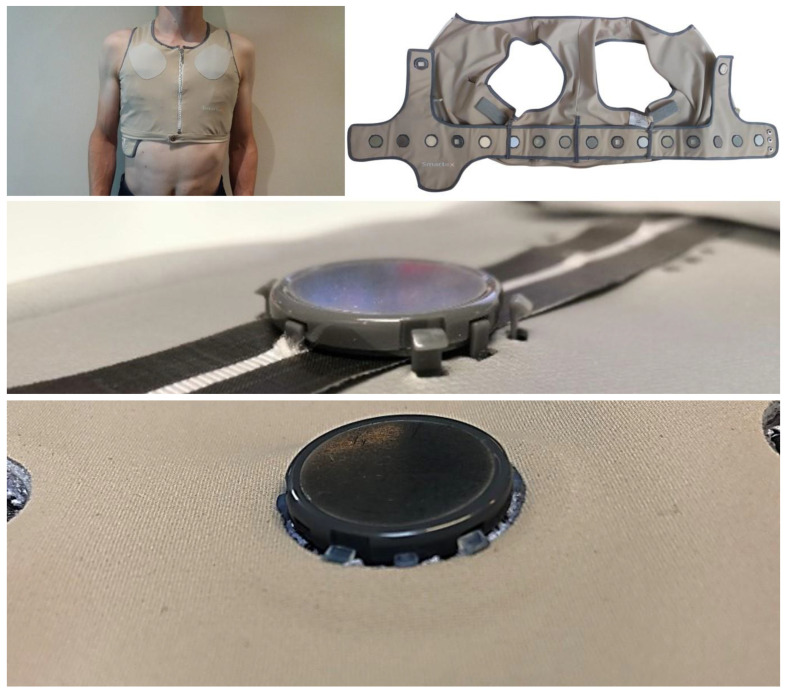
(**Top row**): the ASIC sensor harness worn (**left**) and open (**right**), exposing the sensors and the electrodes on the sensors and two textile electrodes. (**Middle row**): the ASIC sensor clamped to a 3D knit on two electrically conductive tracks, realized as conductive tapes (black). (**Bottom row**): the belt textile with the reinforcement ring (seen as a slight bump in the photo) added on top of the conductive tapes.

**Table 1 sensors-24-05896-t001:** A comparison to existing work (the main contributions of the paper are highlighted in gray).

Technique/Features	Ref.	Comment
Conventional star arrangement		Not suitable for wearables with many electrodes
Passive electrodes, shielded cables	[1]	Widespread
Active electrodes, multi-wire cables	[2,3]	Well known, but not often used
Parallel bus arrangement		Scalable (connector size independent of nb. of electr.)
Bus with more than two wires	[4]	Not easily flexible, stretchable, breathable, washable
Two-wire bus (cooperative sensors)	Section 2	Simplest connection
Locally powered Bootstrapping Separate potential/impedance wires	Section 2.1 Section 2.2 Section 2.2	Easy to comply with safety (medical standards) Suitable for dry electrodes, easy current source Wire impedance is not part of measured bioimpedance
Remotely powered (biopotential only)	[19]	
Remotely powered (+bioimpedance)	Section 3 and Section 4	Sensors can be miniaturized
No monitoring of leakage currents No bootstrapping No separate potential/impedance wires	Section 3.1 and Section 4.1	Requires reliable waterproof double insulation Not ideal for dry electrodes, complex current source Measured bioimpedance including wire impedance
Monitorable leakage currents Bootstrapping Separate potential/impedance wires	Section 3.2 and Section 4.2	Suitably flexible, stretchable, breathable, washable Suitable for dry electrodes, high-end current source Wire impedance is not part of measured bioimpedance

**Table 2 sensors-24-05896-t002:** Definition of current channels. See also Figure 3b.

Channel	Injected Current (100 µA rms, 40 kHz Square Wave)
1	① → ⑦
2	② → ⑧
…	…
10	⑩ → ⑯
11	⑪ → ①
…	…
16	⑯ → ⑥
17–25	unused (yet)

**Table 3 sensors-24-05896-t003:** Measured performances (full scale, FS = [−255, 255] Ω).

*R* (Ω) ±0.02%	Measurement (Ω)	Linearity (% FS)	Noise in 0–2.5 Hz (mΩ rms)
250	249.95	−0.02	32.49
200	200.75	0.30	37.38
150	149.54	−0.18	39.79
100	99.32	−0.27	33.19
50	49.97	−0.01	32.95
0	0.47	0.19	30.55

## Data Availability

Data is contained within the article.

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
