# Peer review of "Remotely Powered Two-Wire Cooperative Sensors for Bioimpedance Imaging Wearables"

_sensors, 2024, doi:10.3390/s24185896_

Round 1
Reviewer 1 Report
Comments and Suggestions for Authors
The manuscript describes a way of carrying out high-channel impedance measurements on the body using active electrodes (cooperative electrodes) with a 2-wire bus for interconection. The bus serves for control, power supply and data transfer. The use of a frequency comb eliminates the need for multiplexing through the electrodes. In principle, a voltage is measured at the electrodes. For the impedance measurement, a sinusoidal current is superimposed and the voltage drop across current source of the electrodes is evaluated.
IQ demodulation with low-pass filtering results in a voltage, which is transmitted as a current via a resistor and is available in the central unit as a voltage across a capacitor.
Two examples are shown, one with discrete components and the other with an ASIC. The measurements on test objects with appropriate calibration showed low noise.
Although designed and implemented as a wearable textile, no data from measurements on the body are shown. Are they not available or not yet appropriate for presenting it?
Specific questions:
1. Is there already experience with measuring on the body, both at rest and in moving subjects?
2. The indication of the resistance over a range of -x ... +x is unusual. Wouldn't it be better to give the absolute value, which should not be a hurdle due to the measuring principle?
3. How sensitive is the measurement to noise or failure of the reference electrode?
4. Could the authors explain the measuring principle a little more clearly before discussing the possibilities of energy harvesting?
Author Response
Comment 0: Although designed and implemented as a wearable textile, no data from measurements on the body are shown. Are they not available or not yet appropriate for presenting it?
Response 0: Thank you for this important comment. There are results from measurements on the body. They are presented in another paper. Its reference has been added.
Comment 1: Is there already experience with measuring on the body, both at rest and in moving subjects?
Response 1: Yes there is, at least for the discrete component version, see response 0 above. For the ASIC version, it is not done yet.
Comment 2: The indication of the resistance over a range of -x ... +x is unusual. Wouldn't it be better to give the absolute value, which should not be a hurdle due to the measuring principle?
Response 2: A resistivity or reactivity EIT image would have all positive values, but before processing the measured values can be anywhere on the complex plane (but in our case, we measure only one axis). The measured values are actually potentials, not impedances, but as these potentials depend on the amount of injected current, it is better to present them as impedance not to be "system dependent".
Comment 3: How sensitive is the measurement to noise or failure of the reference electrode?
Response 3: The noise on the reference electrode is seen as "common mode" and is therefore rejected to the extent of the CMRR of the system. Therefore, its contribution to the measurement noise is negligible. However, a failure of the reference electrode (e.g., if disconnected from the body) will generally shift the electrode potentials beyond the amplifier range and cause saturation and therefore loss of signals of all electrodes. This is a general issue that goes beyond our contribution. In our case, we minimize the risk by using large textile electrodes padded with foam for the reference and neutral electrodes (see â‘ in Figure 9).
Comment 4: Could the authors explain the measuring principle a little more clearly before discussing the possibilities of energy harvesting?
Response 4: Sure, but we are not sure we understand what you would like to have more explained. The measuring principle is exactly the same as the one explained in section 2 for the battery-powered solution. Please, could you tell us more precisely what you would like us to explain?
Reviewer 2 Report
Comments and Suggestions for Authors
The paper addresses an interesting topic. However, the following key aspects should be improved:
1) Clarity: schemes are in general not clear. I recommend to adopt standard symbols for electronics. In particular, the sensing blocks are not clear. Where are voltages measured? In the schemes it not clear which signals are digital (clocks?) and which are analog, please improve the discussion and visualization.
2) Supply: please clarify the role of the remote battery. At the beginning (Fig.2) a battery is presented, but then it disappears and remote switching power is provided. Please improve the discussion of power supply of the nodes.
3) Please clarify how the magnitude and phase of impedance are measured and how the sensing frequency is set.
4) Please clarify if this is the first publication in which the ASIC is presented. If it is the first, please provide more details (fabrication technology, area, power consumption…). If not, pleas
5) Novelty: it seems that all the key innovative ideas were already published and here some examples of application are presented. I suggest to better highlight the novelty and better link the reported examples. Also the reported performance (noise, linearity) should be compared with the state of the art in the literature. I recommend to reference in the introduction a recent example of remote impedance detection system in which the sensing electrodes are also used for tele-powering and communication (https://doi.org/10.1109/TAFE.2024.3409396).
6) Fig. 15 is not clear. Please provide a picture clearly showing the ASIC in the package, with connections and incapsulation (i.e. the inner part of Fig. 18).
Author Response
Comment 1: Clarity: schemes are in general not clear. I recommend to adopt standard symbols for electronics. In particular, the sensing blocks are not clear. Where are voltages measured? In the schemes it not clear which signals are digital (clocks?) and which are analog, please improve the discussion and visualization.
Response 1: Thank you for your comment since we realize after checking with the standard IEC 60617 that the symbol for 'impedance' was wrong. It has been corrected in all figures. We did not identify any other symbol non compliant with this standard, at least for the general symbols. Some symbols have been created to simplify the reading of the figures or when they are usual but come from other standards. All symbols are explained in Figure A1 in appendix. Maybe, a source of lack of clarity is the use in Figures 1 to 6 of functional diagrams in electronic schematics. This is not often seen, but is powerful when some functions are not necessarily implemented with electronic circuits or if different implementations exist (e.g., analogue or digital). Since the purpose of the paper is to explain the principles, abstract representations have been chosen for these figures. However, Figures 8, 10, 13, 14 are closer to conventional electronic schematics since they provide electronic details of specific implementations. In Figures 1 to 10, voltages are "measured" where an arrow is drawn between two conductors (see A5 in Figure A1). For instance, in Figure 4, the measured (i.e., acquired) voltage is v2 which is the voltage between the lower line and the common ground (of the sensor). This voltage v2 is a signal that is further processed via the modulator M (following the arrow to the block M). The modulated signal is then used to control the current of the current source drawn between the two lines. These signals are abstract (functional diagrams) and it is irrelevant if their support is analogue or digital (there are possible implementations for each or a mix of them). In Figure 10, however, the electronics scheme as implemented in our demonstrator is detailed and the digital signals are carried by the conductors connected to the block LOGIC. I guess that the lack of clarity mentioned by the reviewer comes more from Figures 1 to 10. To improve clarity, we highlighted in blue the arrows carrying abstract signals (in the sense of functional diagrams) and the blocks processing signals to avoid confusion with electrical lines. A section in the text has also been added to explain this.
Comment 2: Supply: please clarify the role of the remote battery. At the beginning (Fig.2) a battery is presented, but then it disappears and remote switching power is provided. Please improve the discussion of power supply of the nodes.
Response 2: Figures 2 and 4 show the state of the art of cooperative sensors where each sensor is locally powered by a battery. This is explicitly indicated with the battery symbol, but the indication has to be seen as a 'label', not a battery as an electronic component (since the link from the battery, power supply, to the operation of the two current sources is not detailed). It is the main objective of the paper to show how these batteries can be avoided and replaced by the power supply (usually a battery) available in the central unit. For the same reasons as for the battery before, this power supply is not detailed at the level of the functional diagrams in Figures 5 and 6, but it is visible in the more detailed electronic schematic in Figure 10. Instead, in the Figures 5 and 6, a more abstract symbol is used as the voltage source U (which can be understood in light of Figure 10 as the battery and the switch allowing commutation between 0 and VBAT at 500 Hz). In the functional diagrams in Figures 5 and 6, the local power supply is no more a battery, but the harvesting of energy from U which is functionally equivalent to a current source (whose current multiplied by the voltage U will be the harvested power). This is the reason why the current source symbol is used instead of the battery symbol in the sensors. As the local battery or the harvesting current source have the special function of providing local power supply to the electronic circuits (not shown in the functional diagram in Figure 5, but a bit more detailed in Figure 6 where an op amp is shown (this op amp needs power and this power comes either from the battery in the state of the art or from the harvester in the presented new matter). This op amp is a possible implementation of the current source i1.
Comment 3: Please clarify how the magnitude and phase of impedance are measured and how the sensing frequency is set.
Response 3: Magnitude and phase are not directly measured. Rather (but equivalently) the projections of the vector on specific axes are measured via AM demodulation with a carrier at the phase defining the projection axis. This is shown in Figure 10 where the amplified signal (output of â‘¡) is demodulated (i.e., multiplied,/chopped) by the digital signal controlling the switches (which is generated with the desired phase by the LOGIC). This is the classical IQ demodulation as explained for example in wikipedia (IQ means two orthogonal axes). In Figure 10, the circuit for only one axis is shown.
Comment 4: Please clarify if this is the first publication in which the ASIC is presented. If it is the first, please provide more details (fabrication technology, area, power consumption…). If not, pleas
Response 4: This is indeed the first publication where the ASIC for bioimpedance is published. However, this ASIC is an extension of the ASIC published in the companion paper [14] dealing with biopotential. Details on the technology, area, and power consumption have been added.
Comment 5: Novelty: it seems that all the key innovative ideas were already published and here some examples of application are presented. I suggest to better highlight the novelty and better link the reported examples. Also the reported performance (noise, linearity) should be compared with the state of the art in the literature. I recommend to reference in the introduction a recent example of remote impedance detection system in which the sensing electrodes are also used for tele-powering and communication (https://doi.org/10.1109/TAFE.2024.3409396).
Response 5: Table 1 states the novelty presented in the paper (in grey) with respect of the state of the art (in white). The performances have been discussed and compared with the literature (end of section 4.1.1). Thank you for pointing out the paper 10.1109/TAFE.2024.3409396. However, this paper is too different from what we present to be cited. In this paper, there are four electrodes used to measure the impedance between one of them (in turn) and the others. These four electrodes are used as bus wires (when not used for measurement), two dedicated wires for communication and two for DC powering. The application is for water leakage of pipes. We present an approach for EIT imaging (of bodies) and most importantly (and the difficulty), we use only two wires for powering, communication, and reference potential and current return since our electrodes are in the sensors at any place along the bus. We also have much less space available to have a local supercap as in the suggested paper. As far as we know, our cooperative sensor approach with a 2-wire parallel bus is unique. There are publications (which we cites) presenting a 2-wire parallel bus for powering and communication, but the difficulty we add is the metrology of bioimpedance (and biopotential in a previous paper, we cite also). In our development history, the powering was added last and this is the way we presented it in the paper. In any case, the difficulty is to have all features in the same system (any subset of features is easier).
Comment 6: Fig. 15 is not clear. Please provide a picture clearly showing the ASIC in the package, with connections and incapsulation (i.e. the inner part of Fig. 18).
Response 6: Thank you for the remark. We improved the picture and caption text to clarify.
Round 2
Reviewer 2 Report
Comments and Suggestions for Authors
My comments have been properly addressed.